# Locality-Sensitive Hashing for $f$-Divergences and Kreĭn Kernels: Mutual Information Loss and Beyond

**Lin Chen**[1,2]    **Hossein Esfandiari**[2]    **Thomas Fu**[2]    **Vahab S. Mirrokni**[2]
[1]Yale University    [2]Google Research
lin.chen@yale.edu, {esfandiari,thomasfu,mirrokni}@google.com

## Abstract

Computing approximate nearest neighbors in high dimensional spaces is a central problem in large-scale data mining with a wide range of applications in machine learning and data science. A popular and effective technique in computing nearest neighbors approximately is the *locality-sensitive hashing* (LSH) scheme. In this paper, we aim to develop LSH schemes for distance functions that measure the distance between two probability distributions, particularly for $f$-divergences as well as a generalization to capture mutual information loss. First, we provide a general framework to design LHS schemes for $f$-divergence distance functions and develop LSH schemes for the *generalized Jensen-Shannon divergence* and *triangular discrimination* in this framework. We show a two-sided approximation result for approximation of the generalized Jensen-Shannon divergence by the Hellinger distance, which may be of independent interest. Next, we show a general method of reducing the problem of designing an LSH scheme for a Kreĭn kernel (which can be expressed as the difference of two positive definite kernels) to the problem of maximum inner product search. We exemplify this method by applying it to the *mutual information loss*, due to its several important applications such as model compression.

## 1   Introduction

A central problem in machine learning and data mining is to find top-$k$ similar items to each item in a dataset. Such problems, referred to as *approximate nearest neighbor* problems, are especially challenging in high dimensional spaces and are an important part of a wide range of data mining tasks such as finding near-duplicate pages in a corpus of images or web pages, or clustering items in a high-dimensional metric space. A popular technique for solving these problems is the *locality-sensitive hashing* (LSH) technique [19]. In this method, items in a high-dimensional metric space are first mapped into buckets (via a hashing scheme) with the property that closer items have a higher chance of being assigned to the same bucket. LSH-based nearest neighbor methods limit their scope of search to the items that fall into the same bucket in which the target item resides [1].

Locality sensitive hashing was first introduced and studied by [19]. They provide a family of basic locality-sensitive hash functions for the Hamming distance in a $d$-dimensional space

and for the $L^1$ distance in a $d$-dimensional Euclidean space. They also show that such a family of hash functions provides a randomized $(1 + \epsilon)$-approximation algorithm for the nearest neighbor search problem with sublinear space and sublinear query time. Following [19], several families of locality-sensitive hash functions have been designed and implemented for different metrics, each serving a certain application. We summarize further results in this area in Section 1.1.

In several applications, data points can be represented as probability distributions. One example is the space of users' browsed web pages, read articles or watched videos. In order to represent such data, one can represent each user by a distribution of documents they read, and the documents by topics included in those documents. Other examples are time series distributions, content of documents, or images that can be represented as histograms. Particularly, analysis of similarities in time series distributions or documents can be used in the context of attacks, and spam detection. Analysis of user similarities can be used in recommendation systems and online advertisement.

In fact, many of the aforementioned applications deal with huge datasets and require very time efficient algorithms to find similar data points. These applications motivated us to study LSH functions for distributions, especially for distance measures with information-theoretic justifications. In fact, in addition to $k$-nearest neighbor, LSH functions can be used to implement very fast distributed algorithms for traditional clusterings such as $k$-means [7].

Recently, Mao et al. [26] noticed the importance and lack of LSH functions for the distance of distributions, especially for information-theoretic measures. They attempted to design an LSH to capture the famous *Jensen-Shannon (JS) divergence*. However, instead of directly providing locality-sensitive hash functions for Jensen-Shannon divergence, they take two steps to turn this distance function into a new distance function that is easier to hash. They first looked at a less common divergence measure *S2JSD* which is the square root of two times the JS divergence. Then they defined a related distance function $S2JSD_{new}^{approx}$, which was obtained by only keeping the linear terms in the Taylor expansion of the logarithm in the expression of S2JSD and designed a locality-sensitive hash function for the new measure $S2JSD_{new}^{approx}$. This is an interesting work; however, unfortunately it does not provide any bound on the actual JS divergence using the LSH that they designed for $S2JSD_{new}^{approx}$. Our results resolve this issue by providing LSH schemes with provable guarantees for information-theoretic distance measures including the JS divergence and its generalizations.

Mu and Yan [27] proposed an LSH algorithm for non-metric data by embedding them into a reproducing kernel Kreĭn space. However, their method is indeed data-dependent. Given a finite set of data points $\mathcal{M}$, they compute the distance matrix $D$ whose $(i, j)$-entry is the distance between $i$ and $j$, where both $i$ and $j$ are data points in $\mathcal{M}$. Data is embedded into a reproducing kernel Kreĭn space by performing singular value decomposition on a transform of the distance matrix $D$. The embedding changes if we are given another dataset.

**Our Contributions.** In this paper, we first study LSH schemes for $f$-divergences[2] between two probability distributions. We first in Proposition 1 provide a simple reduction tool for designing LSH schemes for the family of $f$-divergence distance functions. This proposition is not hard to prove but might be of independent interest. Next we use this tool and provide LSH schemes for two examples of $f$-divergence distance functions, Jensen-Shannon divergence and triangular discrimination. Interestingly our result holds for a generalized version of Jensen-Shannon divergence. We apply this tool to design and analyze an LSH scheme for the generalized Jensen-Shannon (GJS) divergence through approximation by the squared Hellinger distance. We use a similar technique to provide an LSH for triangular discrimination. Our approximation is provably lower bounded by a factor 0.69 for the Jensen-Shannon divergence and is lower bounded by a factor 0.5 for triangular discrimination. The approximation result of the generalized Jensen-Shannon divergence by the squared Hellinger requires a more involved analysis and the lower and upper bounds depend on the weight parameter. This approximation result may be of independent interest for other machine learning tasks such as approximate information-theoretic clustering [12]. Our technique may be useful for designing LSH schemes for other $f$-divergences.

Next, we propose a general approach to designing an LSH for Kreĭn kernels. A Kreĭn kernel is a kernel function that can be expressed as the difference of two positive definite kernels. Our approach is built upon a reduction to the problem of maximum inner product search (MIPS) [33, 28, 41]. In contrast to our LSH schemes for $f$-divergence functions via approximation, our approach for Kreĭn kernels involves no approximation and is theoretically *lossless*. Contrary to [27], this approach is data-independent. We exemplify our approach by designing an LSH function specifically for mutual information loss. Mutual information loss is of our particular interest due to its several important applications such as model compression [6, 17], and compression in discrete memoryless channels [20, 30, 42].

## 1.1 Other Related Work

Datar et al. [16] designed an LSH for $L^p$ distances using $p$-stable distributions. Broder [10] designed MinHash for the Jaccard similarity. LSH for other distances and similarity measures were proposed later, for example, angle similarity [11], spherical LSH on a unit hypersphere [34], rank similarity [40], and non-metric LSH [27]. Li et al. [24] demonstrated that uniform quantization outperforms the standard method in [16] with a random offset. Gorisse et al. [18] proposed an LSH family for $\chi^2$ distance by relating it to the $L^2$ distance via an algebraic transform. Interested readers are referred to a more comprehensive survey of existing LSH methods [38]. Another related problem is the construction of feature maps of positive definite kernels. A feature map maps a data point into a usually higher-dimensional space such that the inner product in that space agrees with the kernel in the original space. Explicit feature maps for additive kernels are introduced in [35]. Bregman divergences are another broad class of distances that arise naturally in practical applications. The nearest neighbor search problem for Bregman divergences were studied in [3, 2, 1].

## 2 Preliminaries

### 2.1 Locality-Sensitive Hashing

Let $\mathcal{M}$ be the universal set of items (the database), endowed with a distance function $D$. Ideally, we would like to have a family of hash functions such that for any two items $p$ and $q$ in $\mathcal{M}$ that are close to each other, their hash values collide with a higher probability, and if they reside far apart, their hash values collide with a lower probability. A family of hash functions with the above property is said to be locality-sensitive. A hash value is also known as a bucket in other literature. Using this metaphor, hash functions are imagined as sorters that place items into buckets. If hash functions are locality-sensitive, it suffices to search the bucket into which an item falls if one wants to know its nearest neighbors. The $(r_1, r_2, p_1, p_2)$-sensitive LSH family formulates the intuition of locality sensitivity and is formally defined in Definition 1.

**Definition 1** ([19]). Let $\mathcal{H} = \{h : \mathcal{M} \to U\}$ be a family of hash functions, where $U$ is the set of possible hash values. Assume that there is a distribution $h \sim \mathcal{H}$ over the family of functions. This family $\mathcal{H}$ is called $(r_1, r_2, p_1, p_2)$-sensitive ($r_1 < r_2$ and $p_1 > p_2$) for $D$, if for $\forall p, q \in \mathcal{M}$ the following statements hold: (1) if $D(p, q) \leq r_1$, then $\Pr_{h \sim \mathcal{H}}[h(p) = h(q)] \geq p_1$; (2) if $D(p, q) > r_2$, then $\Pr_{h \sim \mathcal{H}}[h(p) = h(q)] \leq p_2$.

We would like to note that the gap between the high probability $p_1$ and $p_2$ can be amplified by constructing a compound hash function that concatenates multiple functions from an LSH family. For example, one can construct $g : \mathcal{M} \to U^K$ such that $g(p) \triangleq (h_1(p), \ldots, h_K(p))$ for $\forall p \in \mathcal{M}$, where $h_1, \ldots, h_K$ are chosen from the LSH family $\mathcal{H}$. This conjunctive construction reduces the amount of items in one bucket. To improve the recall, an additional disjunction is introduced. To be precise, if $g_1, \ldots, g_L$ are $L$ such compound hash functions, we search all of the buckets $g_1(p), \ldots, g_L(p)$ in order to find the nearest neighbors of $p$.

### 2.2 $f$-Divergence

Let $P$ and $Q$ be two probability measures associated with a common sample space $\Omega$. We write $P \ll Q$ if $P$ is absolutely continuous with respect to $Q$, which requires that for every subset $A$ of $\Omega$, $Q(A) = 0$ imply $P(A) = 0$.

Let $f : (0, \infty) \to \mathbb{R}$ be a convex function that satisfies $f(1) = 0$. If $P \ll Q$, the $f$-divergence from $P$ to $Q$ [14] is defined by

$$D_f(P \parallel Q) = \int_\Omega f\left(\frac{dP}{dQ}\right) dQ, \tag{1}$$

provided that the right-hand side exists, where $\frac{dP}{dQ}$ is the Radon-Nikodym derivative of $P$ with respect to $Q$. In general, an $f$-divergence is not symmetric: $D_f(P \parallel Q) \neq D_f(Q \parallel P)$. If $f_{\mathrm{KL}}(t) = t \ln t + (1-t)$, the $f_{\mathrm{KL}}$-divergence yields the *KL divergence* $D_{\mathrm{KL}}(P \parallel Q) = \int_\Omega \ln \frac{dP}{dQ} dP$ [13]. If $\mathrm{hel}(t) = \frac{1}{2}(\sqrt{t} - 1)^2$, the hel-divergence is the *squared Hellinger distance* $H^2(P, Q) = \frac{1}{2} \int_\Omega (\sqrt{dP} - \sqrt{dQ})^2$ [15]. If $\delta(t) = \frac{(t-1)^2}{t+1}$, the $\delta$-divergence is the *triangular discrimination* (also known as Vincze-Le Cam distance) [22, 36]. If the sample space is finite, the triangular discrimination between $P$ and $Q$ is given by $\Delta(P \parallel Q) = \sum_{i \in \Omega} \frac{(P(i) - Q(i))^2}{P(i) + Q(i)}$.

The *Jensen-Shannon (JS) divergence* is a symmetrized version of the KL divergence. If $P \ll Q$, $Q \ll P$ and $M = (P + Q)/2$, the JS divergence is defined by

$$D_{\mathrm{JS}}(P \parallel Q) = \frac{1}{2} D_{\mathrm{KL}}(P \parallel M) + \frac{1}{2} D_{\mathrm{KL}}(Q \parallel M) \ . \tag{2}$$

## 2.3 Mutual Information Loss and Generalized Jensen-Shannon Divergence

The mutual information loss arises naturally in many machine learning tasks, such as information-theoretic clustering [17] and categorical feature compression [6].

Suppose that two random variables $X$ and $C$ obeys a joint distribution $p(X, C)$. This joint distribution can model a dataset where $X$ denotes the feature value of a data point and $C$ denotes its label [6]. Let $\mathcal{X}$ and $\mathcal{C}$ denote the support of $X$ and $C$ (*i.e.*, the universal set of all possible feature values and labels), respectively. Consider clustering two feature values into a new combined value. This operation can be represented by the following map

$$\pi_{x,y} : \mathcal{X} \to \mathcal{X} \setminus \{x, y\} \cup \{z\} \quad \text{such that} \quad \pi_{x,y}(t) = \begin{cases} t, & t \in \mathcal{X} \setminus \{x, y\} \ , \\ z, & t = x, y \ , \end{cases}$$

where $x$ and $y$ are the two feature values to be clustered and $z \notin \mathcal{X}$ is the new combined feature value. To make the dataset after applying the map $\pi_{x,y}$ preserve as much information of the original dataset as possible, one has to select two feature values $x$ and $y$ such that the mutual information loss incurred by the clustering operation $\mathrm{mil}(x, y) = I(X; C) - I(\pi_{x,y}(X); C)$ is minimized, where $I(\cdot; \cdot)$ is the mutual information between two random variables [13]. Note that the *mutual information loss (MIL) divergence* $\mathrm{mil} : \mathcal{X} \times \mathcal{X} \to \mathbb{R}$ is symmetric in both arguments and always non-negative due to the data processing inequality [13].

Next, we motivate the generalized Jensen-Shannon divergence. If we let $P$ and $Q$ be the conditional distribution of $C$ given $X = x$ and $X = y$, respectively, such that $P(c) = p(C = c|X = x)$ and $Q(c) = p(C = c|X = y)$, the mutual information loss can be re-written as

$$\lambda D_{\mathrm{KL}}(P \parallel M_\lambda) + (1 - \lambda) D_{\mathrm{KL}}(Q \parallel M_\lambda) \ , \tag{3}$$

where $\lambda = \frac{p(x)}{p(x) + p(y)}$ and the distribution $M_\lambda = \lambda P + (1 - \lambda) Q$. Note that (3) is a generalized version of (2). Therefore, we define the *generalized Jensen-Shannon (GJS) divergence* between $P$ and $Q$ [25, 5, 17] by $D_{\mathrm{GJS}}^\lambda(P \parallel Q) = \lambda D_{\mathrm{KL}}(P \parallel M_\lambda) + (1 - \lambda) D_{\mathrm{KL}}(Q \parallel M_\lambda)$, where $\lambda \in [0, 1]$ and $M_\lambda = \lambda P + (1 - \lambda) Q$. We immediately have $D_{\mathrm{GJS}}^{1/2}(P \parallel Q) = D_{\mathrm{JS}}(P \parallel Q)$, which indicates that the JS divergence is indeed a special case of the GJS divergence when $\lambda = 1/2$. The GJS divergence has another equivalent definition $D_{\mathrm{GJS}}^\lambda(P \parallel Q) = H(M_\lambda) - \lambda H(P) - (1 - \lambda) H(Q)$, where $H(\cdot)$ denotes the Shannon entropy [13]. In contrast to the MIL divergence, the GJS $D_{\mathrm{GJS}}^\lambda(\cdot \parallel \cdot)$ is not symmetric in general as the weight $\lambda \in [0, 1]$ is fixed and not necessarily equal to $1/2$. We will show in Lemma 1 that the GJS divergence is an $f$-divergence.

## 2.4 Positive Definite Kernel and Kreĭn Kernel

We first review the definition of a positive definite kernel.

**Definition 2** (Positive definite kernel [32]). Let $\mathcal{X}$ be a non-empty set. A symmetric, real-valued map $k : \mathcal{X} \times \mathcal{X} \to \mathbb{R}$ is a positive definite kernel on $\mathcal{X}$ if for all positive integer $n$, real numbers $a_1, \ldots, a_n \in \mathbb{R}$, and $x, \ldots, x_n \in \mathcal{X}$, it holds that $\sum_{i=1}^{n} \sum_{j=1}^{n} a_i a_j k(x_i, x_j) \geq 0$.

A kernel is said to be a *Kreĭn* kernel if it can be represented as the difference of two positive definite kernels. The formal definition is presented below.

**Definition 3** (Kreĭn kernel [29]). Let $\mathcal{X}$ be a non-empty set. A symmetric, real-valued map $k : \mathcal{X} \times \mathcal{X} \to \mathbb{R}$ is a Kreĭn kernel on $\mathcal{X}$ if there exists two positive definite kernels $k_1$ and $k_2$ on $\mathcal{X}$ such that $k(x, y) = k_1(x, y) - k_2(x, y)$ holds for all $x, y \in \mathcal{X}$.

## 3   LSH Schemes for $f$-Divergences

We build LSH schemes for $f$-divergences based on approximation via another $f$-divergence if the latter admits an LSH family. If $D_f$ and $D_g$ are two divergences associated with convex functions $f$ and $g$ as defined by (1), the approximation ratio of $D_f(P \parallel Q)$ to $D_g(P \parallel Q)$ is determined by the ratio of the functions $f$ and $g$, as well as the ratio of $P$ to $Q$ (to be precise, $\inf_{i \in \Omega} \frac{P(i)}{Q(i)}$) [31].

**Proposition 1** (Proof in Appendix A.4). *Let $\beta_0 \in (0, 1), L, U > 0$ and let $f$ and $g$ be two convex functions $(0, \infty) \to \mathbb{R}$ that obey $f(1) = 0$, $g(1) = 0$, and $f(t), g(t) > 0$ for every $t \neq 1$. Let $\mathcal{P}$ be a set of probability measures on a finite sample space $\Omega$ such that for every $i \in \Omega$ and $P, Q \in \mathcal{P}$, $0 < \beta_0 \leq \frac{P(i)}{Q(i)} \leq \beta_0^{-1}$. Assume that for every $\beta \in (\beta_0, 1) \cup (1, \beta_0^{-1})$, it holds that $0 < L \leq \frac{f(\beta)}{g(\beta)} \leq U < \infty$. If $\mathcal{H}$ forms an $(r_1, r_2, p_1, p_2)$-sensitive family for $g$-divergence on $\mathcal{P}$, then it is also an $(Lr_1, Ur_2, p_1, p_2)$-sensitive family for $f$-divergence on $\mathcal{P}$.*

Proposition 1 provides a general strategy of constructing LSH families for $f$-divergences. The performance of such LSH families depends on the tightness of the approximation. In Sections 3.1 and 3.2, as instances of the general strategy, we derive concrete results for the generalized Jensen-Shannon divergence and triangular discrimination, respectively.

### 3.1   Generalized Jensen-Shannon Divergence

First, Lemma 1 shows that the GJS divergence is indeed an instance of $f$-divergence.

**Lemma 1** (Proof in Appendix A.3). *Define $m_\lambda(t) = \lambda t \ln t - (\lambda t + 1 - \lambda) \ln(\lambda t + 1 - \lambda)$. For any $\lambda \in [0, 1]$, $m_\lambda(t)$ is convex on $(0, \infty)$ and $m_\lambda(1) = 0$. Furthermore, $m_\lambda$-divergence yields the GJS divergence with parameter $\lambda$.*

We choose to approximate it via the squared Hellinger distance, which plays a central role in the construction of the hash family with desired properties.

The approximation guarantee is established in Theorem 1. We show that the ratio of $D_{\mathrm{GJS}}^\lambda(P \parallel Q)$ to $H^2(P, Q)$ is upper bounded by the function $U(\lambda)$ and lower bounded by the function $L(\lambda)$. Furthermore, Theorem 1 shows that $U(\lambda) \leq 1$, which implies that the squared Hellinger distance is an upper bound of the GJS divergence.

**Theorem 1** (Proof in Appendix A.2). *We assume that the sample space $\Omega$ is finite. Let $P$ and $Q$ be two different distributions on $\Omega$. For every $t > 0$ and $\lambda \in (0, 1)$, we have*

$$L(\lambda)H^2(P, Q) \leq D_{\mathrm{GJS}}^\lambda(P \parallel Q) \leq U(\lambda)H^2(P, Q) \leq H^2(P, Q),$$

*where $L(\lambda) = 2\min\{\eta(\lambda), \eta(1 - \lambda)\}$, $\eta(\lambda) = -\lambda \ln \lambda$ and $U(\lambda) = \frac{2\lambda(1-\lambda)}{1-2\lambda} \ln \frac{1-\lambda}{\lambda}$.*

We show Theorem 1 by showing a two-sided approximation result regarding $m_\lambda$ and hel. This result might be of independent interest for other machine learning tasks, say, approximate information-theoretic clustering [12].

**Lemma 2** (Proof in Appendix A.1). *Define $\kappa_\lambda(t) = \frac{m_\lambda(t)}{\mathrm{hel}(t)}$. For every $t > 0$ and $\lambda \in (0, 1)$, we have $\kappa_\lambda(t) = \kappa_{1-\lambda}(1/t)$ and $\kappa_\lambda(t) \in [L(\lambda), U(\lambda)]$.*

We illustrate the upper and lower bound functions $U(\lambda)$ and $L(\lambda)$ in Appendix B. Recall that if $\lambda = 1/2$, the generalized Jensen-Shannon divergence reduces to the usual Jensen-Shannon divergence. Theorem 1 yields the approximation guarantee $0.69 < \ln 2 \leq \frac{D_{\text{JS}}(P\|Q)}{H^2(P,Q)} \leq 1$.

If the common sample space $\Omega$ with which the two distributions $P$ and $Q$ are associated is finite, one can identify $P$ and Q with the $|\Omega|$-dimensional vectors $[P(i)]_{i\in\Omega}$ and $[Q(i)]_{i\in\Omega}$, respectively. In this case, $H^2(P,Q) = \frac{1}{2}\|\sqrt{P} - \sqrt{Q}\|_2^2$, which is exactly half of the squared $L^2$ distance between the two vectors $\sqrt{P} \triangleq [\sqrt{P(i)}]_{i\in\Omega}$ and $\sqrt{Q} \triangleq [\sqrt{Q(i)}]_{i\in\Omega}$. Therefore, the squared Hellinger distance can be endowed with the $L^2$-LSH family [16] applied to the square root of the vector. In light of this, the locality-sensitive hash function that we propose for the generalized Jensen-Shannon divergence is

$$h_{\mathbf{a},b}(P) = \left\lceil \frac{\mathbf{a} \cdot \sqrt{P} + b}{r} \right\rceil, \tag{4}$$

where $\mathbf{a} \sim \mathcal{N}(0, I)$ is a $|\Omega|$-dimensional standard normal random vector, $\cdot$ denotes the inner product, $b$ is uniformly at random on $[0, r]$, and $r$ is a positive real number.

**Theorem 2** (Proof in Appendix A.5). *Let $c = \|\sqrt{P} - \sqrt{Q}\|_2$ and $f_2$ be the probability density function of the absolute value of the standard normal distribution. The hash functions $\{h_{\mathbf{a},b}\}$ defined in (4) form a $(R, c^2 \frac{U(\lambda)}{L(\lambda)} R, p_1, p_2)$-sensitive family for the generalized Jensen-Shannon divergence with parameter $\lambda$, where $R > 0$, $p_1 = p(1)$, $p_2 = p(c)$, and $p(u) = \int_0^r \frac{1}{u} f_2(t/u)(1 - t/r)dt$.*

## 3.2 Triangular Discrimination

Recall that triangular discrimination is the $\delta$-divergence, where $\delta(t) = \frac{(t-1)^2}{t+1}$. As shown in the proof of Theorem 3 (Appendix A.6), the function $\delta$ can be approximated by the function $\text{hel}(t)$ that defines the squared Hellinger distance $1 \leq \frac{\delta(t)}{\text{hel}(t)} \leq 2$. The squared Hellinger distance can be sketched via $L^2$-LSH after taking the square root, as exemplified in Section 3.1. By Proposition 1, the LSH family for the square Hellinger distance also forms an LSH family for the triangular discrimination. Theorem 3 shows that the LSH family defined in (4) form a $(R, 2c^2 R, p_1, p_2)$-sensitive family for triangular discrimination.

**Theorem 3** (Proof in Appendix A.6). *Let $c = \|\sqrt{P} - \sqrt{Q}\|_2$ and $f_2$ be the probability density function of the absolute value of the standard normal distribution. The hash functions $\{h_{\mathbf{a},b}\}$ defined in (4) form a $(R, 2c^2 R, p_1, p_2)$-sensitive family for triangular discrimination, where $R > 0$, $p_1 = p(1)$, $p_2 = p(c)$, and $p(u) = \int_0^r \frac{1}{u} f_2(t/u)(1 - t/r)dt$.*

# 4 Kreĭn-LSH for Mutual Information Loss

In this section, we first show that the mutual information loss is a Kreĭn kernel. Then we propose *Kreĭn-LSH*, an asymmetric LSH method [33] for mutual information loss. We would like to remark that this method can be easily extended to other Kreĭn kernels, provided that the associated positive definite kernels allow an explicit feature map.

## 4.1 Mutual Information Loss is a Kreĭn Kernel

Recall that in Section 2.3 we assume a joint distribution $p(X, C)$ whose support is $\mathcal{X} \times \mathcal{C}$. Let $x, y \in \mathcal{X}$ be represented by $\mathbf{x} = [p(c, x) : c \in \mathcal{C}] \in [0, 1]^{|\mathcal{C}|}$ and $\mathbf{y} = [p(c, y) : c \in \mathcal{C}] \in [0, 1]^{|\mathcal{C}|}$, respectively. We consider the mutual information loss of merging $x$ and $y$, which is given by $I(X; C) - I(\pi_{x,y}(X); C)$.

**Theorem 4** (Proof in Appendix A.8). *The mutual information loss* $\text{mil}(\mathbf{x}, \mathbf{y})$ *is a Kreĭn kernel on* $[0, 1]^{|\mathcal{C}|}$. *In other words, there exist two positive definite kernels* $K_1$ *and* $K_2$ *on* $[0, 1]^{|\mathcal{C}|}$ *such that* $\text{mil}(\mathbf{x}, \mathbf{y}) = K_1(\mathbf{x}, \mathbf{y}) - K_2(\mathbf{x}, \mathbf{y})$. *To be explicit, we set* $K_1(\mathbf{x}, \mathbf{y}) = k(\sum_{c\in\mathcal{C}} p(c, x), \sum_{c\in\mathcal{C}} p(c, y))$ *and* $K_2(\mathbf{x}, \mathbf{y}) = \sum_{c\in\mathcal{C}} k(p(c, x), p(c, y))$, *where* $k(a, b) = a \ln \frac{a}{a+b} + b \ln \frac{b}{a+b}$.

To prove Theorem 4 and construct explicit feature maps for $K_1$ and $K_2$, we need the following lemma.

**Lemma 3** (Proof in Appendix A.7)**.** *The kernel $k$ is a positive definite kernel on $[0, 1]$. Moreover, it is endowed with the following explicit feature map $x \mapsto \Phi_w(x)$ such that $k(x, y) = \int_{\mathbb{R}} \Phi_w(x)^* \Phi_w(y) dw$, where $\Phi_w(x) \triangleq e^{-iw \ln(x)} \sqrt{x \frac{2 \operatorname{sech}(\pi w)}{1 + 4w^2}}$ and $\Phi_w(x)^*$ denotes the complex conjugate of $\Phi_w(x)$.*

The map $\Phi(x) : w \mapsto \Phi_w(x)$ is called the *feature map* of $x$. The integral $\int_{\mathbb{R}} \Phi_w(x)^* \Phi_w(y) dw$ is also denoted by a Hermitian inner product $\langle \Phi(x), \Phi(y) \rangle$.

## 4.2 Kreĭn-LSH for Mutual Information Loss

Now we are ready to present an asymmetric LSH scheme [33] for mutual information loss. This method can be easily extended to other Kreĭn kernels, provided that the associated positive definite kernels admit an explicit feature map. In fact, we reduce the problem of designing the LSH for a Kreĭn kernel to the problem of designing the LSH for maximum inner product search (MIPS) [33, 28, 41]. We call this general reduction *Kreĭn-LSH*.

### 4.2.1 Reduction to Maximum Inner Product Search

Our reduction is based on the following observation. Suppose that $K$ is a Kreĭn kernel on $\mathcal{X}$ such that $K = K_1 - K_2$ where $K_1$ and $K_2$ are positive definite kernels on $\mathcal{X}$. Assume that $K_1$ and $K_2$ admit feature maps $\Phi_1$ and $\Phi_2$ such that $K_1(x, y) = \langle \Psi_1(x), \Psi_1(y) \rangle$ and $K_2(x, y) = \langle \Psi_2(x), \Psi_2(y) \rangle$. Then the Kreĭn kernel $K$ can also represented as an inner product

$$K(x, y) = \langle \Phi_1(x) \oplus \Phi_2(x), \Phi_1(y) \oplus -\Phi_2(y) \rangle \ , \tag{5}$$

where $\oplus$ denotes the direct sum. If we define a pair of transforms $T_1(x) \triangleq \Phi_1(x) \oplus \Phi_2(x)$ and $T_2(x) \triangleq \Phi_1(x) \oplus -\Phi_2(x)$, then we have $K(x, y) = \langle T_1(x), T_2(y) \rangle$. We call this pair of transforms *left and right Kreĭn transforms*.

---

**Algorithm 1** Kreĭn-LSH

---

**Input:** Discretization parameters $J \in \mathbb{N}$ and $\Delta > 0$.
**Output:** The left and right Kreĭn transform $\eta_1$ and $\eta_2$.
  1: $w_j \leftarrow (j - 1/2)\Delta$ for $j = 1, \dots, J$
  2: Construct the atomic transform

$$\tau(x, w, j) \triangleq \left[ \cos(w \ln(x)) \sqrt{2x \int_{(j-1)\Delta}^{j\Delta} \rho(w') dw'}, \sin(w \ln(x)) \sqrt{2x \int_{(j-1)\Delta}^{j\Delta} \rho(w') dw'} \right] \ .$$

  3: Construct the left and right basic transform

$$\eta_1(\mathbf{x}) \triangleq \bigoplus_{j=1}^{J} \tau(p(x), w_j, j) \oplus \bigoplus_{j=1}^{J} \bigoplus_{c \in \mathcal{C}} \tau(p(c, x), w_j, j) \, ,$$

$$\eta_2(\mathbf{x}) \triangleq \bigoplus_{j=1}^{J} \tau(p(x), w_j, j) \oplus \bigoplus_{j=1}^{J} \bigoplus_{c \in \mathcal{C}} -\tau(p(c, x), w_j, j) \, .$$

  4: Construct the left and right Kreĭn transform

$$T_1(\mathbf{x}, M) \triangleq [\eta_1, \sqrt{M - \|\eta_1(\mathbf{x})\|_2^2}, 0], \quad T_2(\mathbf{y}, M) \triangleq [\eta_2, 0, \sqrt{M - \|\eta_2(\mathbf{x})\|_2^2}] \, .$$

    where $M$ is a constant such that $M \geq \|\eta_1(\mathbf{x})\|_2^2$ (note that $\|\eta_1(\mathbf{x})\|_2 = \|\eta_2(\mathbf{x})\|_2$).
  5: Sample $\mathbf{a} \sim \mathcal{N}(0, I)$ and construct the hash function $h(\mathbf{x}; M) \triangleq \operatorname{sign}(\mathbf{a}^\top T(\mathbf{x}, M))$, where $T$ is either the left or right transform.

---

We exemplify this technique by applying it to the MIL divergence. For ease of exposition, we define $\rho(w) \triangleq \frac{2 \operatorname{sech}(\pi w)}{1 + 4w^2}$. The proposed approach Kreĭn-LSH is presented in Algorithm 1.

To make the intuition of (5) applicable in a practical implementation, we have to truncate and discretize the integral $k(x,y) = \int_R \Phi_w(x)^* \Phi_w(y) dw$. First we analyze the truncation. The analysis is similar to Lemma 10 of [4].

**Lemma 4** (Truncation error bound, proof in Appendix A.9). *If $t > 0$ and $x, y \in [0,1]$, the truncation error can be bounded as follows $\left| k(x,y) - \int_{-t}^{t} \Phi_w(x)^* \Phi_w(y) dw \right| \leq 4e^{-t}$.*

To discretize the finite integral $\int_{-t}^{t} \Phi_w(x)^* \Phi_w(y) dw$, we divide the inteval into $2J$ sub-intervals of length $\Delta$. The following lemma bounds the discretization error.

**Lemma 5** (Discretization error bound, proof in Appendix A.10). *If $J$ is a positive integer, $\Delta > 0$, and $w_j = (j - 1/2)\Delta$, the discretization error is bounded as follows $\left| \int_{-\Delta J}^{\Delta J} \Phi_w(x)^* \Phi_w(y) dw - \left\langle \bigoplus_{j=1}^{J} \tau(x, w_j, j), \bigoplus_{j=1}^{J} \tau(y, w_j, j) \right\rangle \right| \leq 2\Delta$, where $\tau(x, w, j) = \left[ \cos(w \ln(x)) \sqrt{2x \int_{(j-1)\Delta}^{j\Delta} \rho(w') dw'}, \sin(w \ln(x)) \sqrt{2x \int_{(j-1)\Delta}^{j\Delta} \rho(w') dw'} \right] \in \mathbb{R}^2$.*

By Lemmas 4 and 5, to guarantee that the total approximation error (including both truncation and discretization errors) is at most $\epsilon$, it suffices to set $\Delta = \frac{\epsilon}{4(1+|\mathcal{C}|)}$ and $J \geq \frac{4(1+|\mathcal{C}|)}{\epsilon} \ln \frac{8(1+|\mathcal{C}|)}{\epsilon}$.

#### 4.2.2  LSH for Maximum Inner Product Search

The second stage of our proposed method is to apply LSH to the MIPS problem. As an example, in Line 5, we use the SIMPLE-LSH introduced by [28]. Let us have a quick review of SIMPLE-LSH. Assume that $\mathcal{M} \subseteq \mathbb{R}^d$ is a finite set of vectors and that for all $\mathbf{x} \in \mathcal{M}$, there is a universal bound on the squared 2-norm, *i.e.*, $\|\mathbf{x}\|_2^2 \leq M$. Neyshabur and Srebro [28] assume that $M = 1$ without loss of generality. We allow $M$ to be any positive real number. For two vectors $\mathbf{x}, \mathbf{y} \in \mathcal{M}$, SIMPLE-LSH performs the following transform $L_1(\mathbf{x}) \triangleq [\mathbf{x}, \sqrt{M - \|\mathbf{x}\|_2^2}, 0], L_2(\mathbf{y}) \triangleq [\mathbf{y}, 0, \sqrt{M - \|\mathbf{y}\|_2^2}]$. Note that the norm of $L_1$ and $L_2$ is $M$ and that therefore their cosine similarity equals their inner product. In fact, SIMPLE-LSH is a reduction from MIPS to LSH for the cosine similarity. Then a random-projection-based LSH for the cosine similarity [11, 38]

$$h(\mathbf{x}) \triangleq \text{sign}(\mathbf{x}^\top L_i(\mathbf{x})), \quad \mathbf{a} \sim \mathcal{N}(0, I), i = 1, 2$$

can be used for MIPS and thereby LSH for the MIL divergence via our reduction.

**Discussion**  We have some important remarks for practical implementation of Kreĭn-LSH. Although [28] provides a theoretical guarantee for LSH for MIPS, as noted in [41], the additional term $\sqrt{M - \|\mathbf{x}\|_2^2}$ may dominate in the 2-norm and significantly degrade the performance of LSH. To circumvent this issue, we recommend a method that partitions the dataset according to the 2-norm, *e.g.*, the norm-ranging method [41].

## 5  Experiment Results

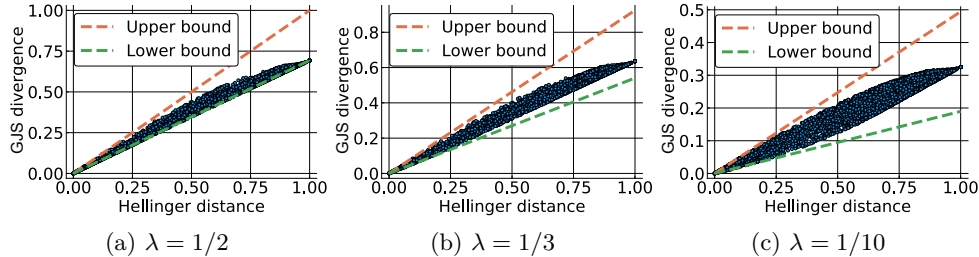

(a) $\lambda = 1/2$  (b) $\lambda = 1/3$  (c) $\lambda = 1/10$

Figure 1: The empirical performance of Hellinger approximation

**Approximation Guarantee.** In the first part, we verify the theoretical bounds derived in Theorem 1 on real data. We used the latent Dirichlet allocation to extract the topic

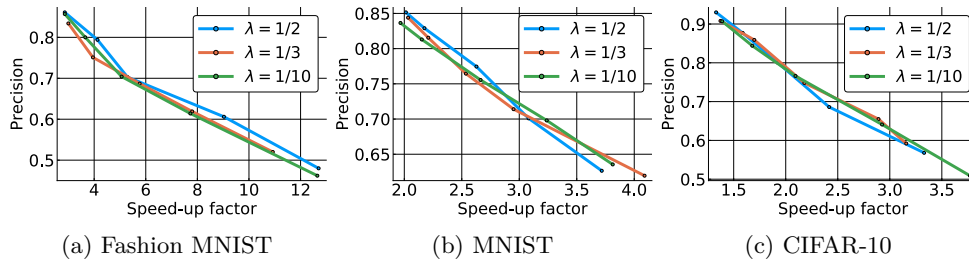

|  |  |  |
|---|---|---|
| (a) Fashion MNIST | (b) MNIST | (c) CIFAR-10 |

Figure 2: Precision vs. speed-up factor for different $\lambda$'s.

distributions of Reuters-21578, Distribution 1.0. The number of topics is set to 10. We sampled 100 documents uniformly at random and computed the GJS divergence and Hellinger distance between each pair of topic distributions. Each dot in Fig. 1 represents the topic distribution of a document. The horizontal axis denotes the Hellinger distance while the vertical axis denotes the GJS divergence. We chose different parameter values ($\lambda = 1/2, 1/3, 1/10$) for the GJS divergence. From the three subfigures, we observe that both the upper and lower bounds are tight for the data.

**Nearest Neighbor Search.** In the second part, we apply the proposed LSH scheme for the GJS divergence to the nearest neighbor search problem in Fashion MNIST [39], MNIST [23], and CIFAR-10 [21]. Each image in the datasets is flattened into a vector and $L^1$-normalized, thereby summing to 1. As described in Section 2.1, a concatenation of hash functions is used. We denote the number of concatenated hash functions by $K$ and the number of compound hash functions by $L$. In the first set of experiments, we set $K = 3$ and vary $L$ from 20 to 40. We measure the execution time of LSH-based $k$-nearest neighbor search and the exact (brute-force) algorithm, where $k$ is set to 20. Both algorithms were run on a 2.2 GHz Intel Core i7 processor. The speed-up factor is the ratio of the execution time of the exact algorithm to that of the LSH-based method. The quality of the result returned by the LSH-based method is quantified by its precision, which is the fraction of correct nearest neighbors among the retrieved items. We would like to remark that the precision and recall are equal in our case since both algorithms return $k$ items. We also vary the parameter of the GJS divergence and choose $\lambda$ from $\{1/2, 1/3, 1/10\}$. The result is illustrated in Figs. 2a to 2c. We observe a trade-off between the quality of the output (precision) and computational efficiency (speed-up factor). The performance appears to be robust to the parameter of the GJS divergence. In the second set of experiments, we fix the parameter of the GJS divergence to $1/2$; *i.e.*, the JS divergence is used. The number of concatenated hash functions $K$ ranges from 3 to 5 or 4 to 6. The result is presented in Appendix C. In addition to the aforementioned quality-efficiency trade-off, we observe that a larger $K$ results in a more efficient algorithm given the same target precision.

## 6 Conclusion

In this paper, we propose a general strategy of designing an LSH family for $f$-divergences. We exemplify this strategy by developing LSH schemes for the generalized Jensen-Shannon divergence and triangular discrimination in this framework. They are endowed with an LSH family via the Hellinger approximation. In particular, we show a two-sided approximation for the generalized Jensen-Shannon divergence by the Hellinger distance. This may be of independent interest. Next, we propose a general approach to designing an LSH scheme for Kreĭn kernels via a reduction to the problem of maximum inner product search. In contrast to our strategy for $f$-divergences, this approach involves no approximation and is theoretically lossless. We exemplify this approach by applying to mutual information loss.

**Acknowledgments**

LC was supported by the Google PhD Fellowship.

## Footnotes

[1]We note that LSH is a popular data-independent technique for nearest neighbor search. Another category of nearest neighbor search algorithms, referred to as data-dependent techniques, are *learning-to-hash* methods [37] which learn a hash function that maps each item to a compact code. However, this line of work is out of the scope of this paper.

[2]The formal definition of $f$-divergence is presented in Section 2.2.

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
