[Supplementary Material]

# Appendix A    Proofs

## A.1    Proof of Lemma 2

The first equation $\kappa_\lambda(t) = \kappa_{1-\lambda}(1/t)$ can be verified directly by plugging in $1 - \lambda$ and $1/t$. In the sequel, we show the second equation $\kappa_\lambda(t) \in [L(\lambda), U(\lambda)]$, which needs a detailed and careful analysis and discussion. The derivative of $\kappa_\lambda$, denoted by $\kappa'_\lambda(t)$, is

$$\frac{2\left(\lambda\left(\sqrt{t} - 1\right) + 1\right)\ln(\lambda(t-1) + 1) - 2\lambda\sqrt{t}\ln(t)}{\left(\sqrt{t} - 1\right)^3\sqrt{t}}.$$

We define $f_1(t) = 2\left(\lambda\left(\sqrt{t} - 1\right) + 1\right)\ln(\lambda(t-1) + 1) - 2\lambda\sqrt{t}\ln(t)$. Its derivative $f'_1(t)$ is

$$-\frac{\lambda}{\sqrt{t}(\lambda(t-1) + 1)}\left(2(\lambda - 1)\left(\sqrt{t} - 1\right) + (\lambda(t-1) + 1)(\log(t) - \log(\lambda(t-1) + 1))\right).$$

Define $f_2(t) = 2(\lambda - 1)\left(\sqrt{t} - 1\right) + (\lambda(t-1) + 1)(\log(t) - \log(\lambda(t-1) + 1))$. Its derivative $f'_2(t)$ is

$$\frac{(\lambda - 1)\left(\sqrt{t} - 1\right)}{t} + \lambda(\log(t) - \log(\lambda(t-1) + 1)).$$

Its second derivative $f''_2(t)$ is

$$\frac{(1 - \lambda)\left(2(\lambda - 1) + \sqrt{t}(\lambda(t-1) + 1)\right)}{2t^2(\lambda(t-1) + 1)}.$$

First, we assume $\lambda \in (0, 1/2)$. In this case, we have $\frac{1-\lambda}{\lambda} > 1$ and $\lambda(t-1) + 1 > 0$. Notice that $f_3(t) = 2(\lambda - 1) + \sqrt{t}(\lambda(t-1) + 1)$ is a strictly increasing function in $t$. Therefore, if $t > \left(\frac{1-\lambda}{\lambda}\right)^2$, we obtain

$$f_3(t) > f_3\left(\left(\frac{1-\lambda}{\lambda}\right)^2\right) = \frac{(\lambda - 1)(\lambda + 1)(2\lambda - 1)}{\lambda^2} > 0.$$

Therefore $f''_2(t) > 0$ if $t > \left(\frac{1-\lambda}{\lambda}\right)^2$. Thus we deduce that $f'_2(t)$ is increasing in $t$ if $t > \left(\frac{1-\lambda}{\lambda}\right)^2$, which yields

$$f'_2(t) > f'_2\left(\left(\frac{1-\lambda}{\lambda}\right)^2\right) = \frac{\lambda\left(2\lambda + (1 - \lambda)\log\left(\frac{1-\lambda}{\lambda}\right) - 1\right)}{1 - \lambda}.$$

Define $g(\lambda) = 2\lambda + (1 - \lambda)\log\left(\frac{1-\lambda}{\lambda}\right) - 1$. Its derivative $g'(\lambda) = -\frac{1}{\lambda} - \log\left(\frac{1}{\lambda} - 1\right) + 2$ is negative if $\lambda < 1/2$ and positive if $\lambda > 1/2$. Therefore $g(\lambda) \geq g(1/2) = 0$. Thus we obtain that if $t > \left(\frac{1-\lambda}{\lambda}\right)^2$, $f'_2(t) > 0$, which implies that $f_2(t)$ is increasing in $t$ if $t > \left(\frac{1-\lambda}{\lambda}\right)^2$. Thus we have

$$f_2(t) > f_2\left(\left(\frac{1-\lambda}{\lambda}\right)^2\right) = \frac{(1 - \lambda)\left(4\lambda + \log\left(\frac{1}{\lambda} - 1\right) - 2\right)}{\lambda}.$$

Define $g_1(\lambda) = 4\lambda + \log\left(\frac{1}{\lambda} - 1\right) - 2$. Its derivative $g'_1(t) = \frac{1}{(\lambda-1)\lambda} + 4$ is non-positive, which implies that $g_1$ is decreasing in $\lambda$. Therefore, if $t > \left(\frac{1-\lambda}{\lambda}\right)^2$, we have

$$f_2(t) > \frac{1 - \lambda}{\lambda}g(1/2) = 0.$$

Since $\lambda(t-1) + 1 > 0$, we obtain that $f'_1(t) < 0$ and therefore $f_1(t)$ is decreasing if $t > \left(\frac{1-\lambda}{\lambda}\right)^2$. We have

$$f_1(t) < f_1\left(\left(\frac{1-\lambda}{\lambda}\right)^2\right) = 0.$$

If $t > \left(\frac{1-\lambda}{\lambda}\right)^2$, since $\left(\sqrt{t} - 1\right)^3\sqrt{t} > 0$, we deduce that $\kappa'_\lambda(t) < 0$.

If $t < 1$, since $f_3(t)$ is strictly increasing in $t$, we have $f_3(t) < f_3(1) = 2\lambda - 1 < 0$, which implies that $f_2''(t) < 0$. Therefore, we obtain that $f_2'(t)$ is strictly decreasing on $(0, 1)$. Thus we have $f_2'(t) > f_2'(1) = 0$, which implies that $f_2(t)$ is strictly increasing on $(0, 1)$. We immediately have $f_2(t) < f_2(1) = 0$ for $\forall t \in (0, 1)$, which yields that $f_1'(t) > 0$ and therefore $f_1(t)$ is strictly increasing on $(0, 1)$. For $\forall t \in (0, 1)$, it holds that $f_1(t) < f_1(1) = 0$. Since $\left(\sqrt{t} - 1\right)^3 \sqrt{t} < 0$, we deduce that $\kappa_\lambda'(t) > 0$ for $t \in (0, 1)$.

The interval that remains unexplored is $I = \left(1, \left(\frac{1-\lambda}{\lambda}\right)^2\right)$. Since $f_3(1) = 2\lambda - 1 < 0$ and $f_3\left(\left(\frac{1-\lambda}{\lambda}\right)^2\right) = \frac{(\lambda-1)(\lambda+1)(2\lambda-1)}{\lambda^2} > 0$, we know that $f_3(t)$ has a real root on this interval. Notice that $f_3(t)$ can be viewed as a cubic function in $\sqrt{t}$. Define $f_4(x) = 2\lambda + \lambda x^3 + (1-\lambda)x - 2$ and we have $f_3(t) = f_4(\sqrt{t})$. The cubic function $f_4$ is strictly monotone if $\lambda \in (0, 1)$. Therefore, the real root of $f_3$ on $I$ is unique and we denote it by $\rho(\lambda)$.

Now we divide the interval $I = \left(1, \left(\frac{1-\lambda}{\lambda}\right)^2\right)$ into two subintervals $I_1 = (1, \rho(\lambda))$ and $I_2 = \left(\rho(\lambda), \left(\frac{1-\lambda}{\lambda}\right)^2\right)$. Since $f_3(t) < 0$ on $I_1$ and $f_3(t) > 0$ on $I_2$, we have $f_2''(t) < 0$ on $I_1$ and $f_2''(t) > 0$ on $I_2$. Therefore, we deduce that $f_2'(t)$ strictly decreases on $I_1$ and strictly increases on $I_2$. Note that $f_2'(1) = 0$ and

$$f_2'\left(\left(\frac{1-\lambda}{\lambda}\right)^2\right) = \frac{\lambda\left(2\lambda + (1-\lambda)\log\left(\frac{1-\lambda}{\lambda}\right) - 1\right)}{1-\lambda} > 0.$$

To see this, we define $g_2(\lambda) = 2\lambda + (1-\lambda)\log\left(\frac{1-\lambda}{\lambda}\right) - 1$. Its second derivative is $g_2''(\lambda) = \frac{1}{\lambda^2 - \lambda^3} > 0$, which implies that $g_2(\lambda)$ is strictly convex and $g_2'(\lambda)$ has a unique root. Observe that $\lambda = 1/2$ is a root of $g_2'(\lambda)$. We deduce that $g_2(\lambda) > g_2(1/2) = 0$ for $\lambda \in (0, 1/2)$, which immediately yields that $f_2'\left(\left(\frac{1-\lambda}{\lambda}\right)^2\right) > 0$. Thus the function $f_2'(t)$ has a unique root (denoted by $\rho_1(\lambda)$) on $I$. Therefore, the function $f_2(t)$ strictly decreases on $I_3 = (1, \rho_1(\lambda))$ and strictly increases on $I_4 = \left(\rho_1(\lambda), \left(\frac{1-\lambda}{\lambda}\right)^2\right)$. Note that $f_2(1) = 0$ and

$$f_2\left(\left(\frac{1-\lambda}{\lambda}\right)^2\right) = \frac{(1-\lambda)\left(4\lambda + \log\left(\frac{1-\lambda}{\lambda}\right) - 2\right)}{\lambda} > 0.$$

To see the above inequality, we define $g_3(\lambda) = 4\lambda + \log\left(\frac{1-\lambda}{\lambda}\right) - 2$. Its derivative is $g_3'(\lambda) = \frac{(1-2\lambda)^2}{(\lambda-1)\lambda} < 0$, which implies that $g_3(\lambda)$ strictly decreases and that $g_3(\lambda) > g_3(1/2) = 0$ for $\lambda \in (0, 1/2)$. As a result, we deduce that $f_2\left(\left(\frac{1-\lambda}{\lambda}\right)^2\right) > 0$. Thus we obtain that the function $f_2(t)$ has a unique root (denoted by $\rho_2(\lambda)$) on $I$ and that $f_1'(t)$ is positive on $I_5 = (1, \rho_2(\lambda))$ and negative on $I_6 = \left(\rho_2(\lambda), \left(\frac{1-\lambda}{\lambda}\right)^2\right)$, which implies that $f_1$ strictly increases on $I_5$ and strictly decreases on $I_6$. Note that $f_1(1) = f_1\left(\left(\frac{1-\lambda}{\lambda}\right)^2\right) = 0$. We conclude that $f_1(t) > 0$ on $I$, which implies that $\kappa_\lambda'(t) > 0$ on $I$.

From the above analysis, we see that if $\lambda \in (0, 1/2)$, the function $\kappa_\lambda'(t)$ has no real root on $(0, \infty) \setminus \{1, \left(\frac{1-\lambda}{\lambda}\right)^2\}$. Since

$$\lim_{t \to 1} \kappa_\lambda(t) = 4(1-\lambda)\lambda > 0, \quad \kappa_\lambda\left(\left(\frac{1-\lambda}{\lambda}\right)^2\right) = 0,$$

we deduce that the derivative $\kappa_\lambda'(t)$ has a unique root at $t = \left(\frac{1-\lambda}{\lambda}\right)^2$ if $\lambda \in (0, 1/2)$. By (**??**), we know that it also holds for $\lambda \in (1/2, 1)$. Furthermore, we know that the derivative is positive if $t < \left(\frac{1-\lambda}{\lambda}\right)^2$ and is negative if $t > \left(\frac{1-\lambda}{\lambda}\right)^2$. Thus the maximum of $\kappa_\lambda$ is attained at $t = \left(\frac{1-\lambda}{\lambda}\right)^2$ and it is exactly $U(\lambda)$.

Next, we assume $\lambda = 1/2$. We have

$$\kappa_{1/2}(t) = \frac{t\log(t) + (t+1)(\log(2) - \log(t+1))}{\left(\sqrt{t} - 1\right)^2}.$$

Its derivative is

$$\kappa'_{1/2}(t) = \frac{\left(\sqrt{t} + 1\right) \log\left(\frac{t+1}{2}\right) - \sqrt{t}\log(t)}{\left(\sqrt{t} - 1\right)^3 \sqrt{t}}$$

Define $f_5(t) = \left(\sqrt{t} + 1\right) \log\left(\frac{t+1}{2}\right) - \sqrt{t}\log(t)$. Its derivative is

$$f'_5(t) = \frac{2\left(\sqrt{t} - 1\right) + (t+1)\log\left(\frac{t+1}{2}\right) - (t+1)\log(t)}{2\sqrt{t}(t+1)}.$$

Then we define $f_6(t) = 2\left(\sqrt{t} - 1\right) + (t+1)\log\left(\frac{t+1}{2}\right) - (t+1)\log(t)$, whose derivative is

$$f'_6(t) = \frac{\sqrt{t} - 1}{t} - \log(2t) + \log(t+1)$$

and second derivative

$$f''_6(t) = \frac{1}{t^3 + t^2} - \frac{1}{2t^{3/2}}.$$

If we set $f''_6(t) > 0$, we get $t^{1/2} + t^{3/2} < 2$, which is equivalent to $t < 1$. Therefore $f''_6(t)$ is positive on $(0,1)$ and negative on $(1,\infty)$, which implies that $f'_6(t) < f'_6(1) = 0$ for $t \neq 1$. We deduce that $f_6(t)$ is strictly decreasing in $t$ and thus has a unique root. Since $t = 1$ is a root of $f_6(t)$, it is the unique root, which implies that $f_6(t)$ and $f'_5(t)$ are both positive on $(0,1)$ and negative on $(1,\infty)$. As a result, we deduce that $f_5(t) < f_5(1) = 0$ for $t \neq 1$. Thus we conclude that $\kappa'_{1/2}(t)$ is positive on $(0,1)$ and negative on $(1,\infty)$. We can verify that $t = 1$ is indeed a root of $\kappa'_{1/2}(t)$.

So far we have shown for $t \in (0,1)$ that the derivative $\kappa'_\lambda(t)$ is positive if $t < \left(\frac{1-\lambda}{\lambda}\right)^2$ and is negative if $t > \left(\frac{1-\lambda}{\lambda}\right)^2$. Thus the maximum of $\kappa_\lambda$ is attained at $t = \left(\frac{1-\lambda}{\lambda}\right)^2$ and it is exactly $U(\lambda)$.

The infimum is

$$\min\{\lim_{t\to 0^+} \kappa_\lambda(t), \lim_{t\to\infty} \kappa_\lambda(t)$$
$$= \min\{-2(1-\lambda)\ln(1-\lambda), -2\lambda\ln\lambda\}.\}$$

Therefore we conclude $\kappa_\lambda \in [L(\lambda), U(\lambda)]$.

### A.2 Proof of Theorem 1

In addition to Lemma 2, we need the following lemma.

**Lemma 6** (Theorem 6 of [31]). *Let $f$ and $g$ be two convex functions that satisfy $f(1) = 0$ and $g(1) = 0$, respectively. The function $g(t) > 0$ for every $t \in (0,1) \cup (1,\infty)$. Let $P$ and $Q$ be two distributions on a common finite sample space $\Omega$. Define $\beta_1 = \inf_{i\in\Omega} \frac{Q(i)}{P(i)}$ and $\beta_2 = \inf_{i\in\Omega} \frac{P(i)}{Q(i)}$. We assume that $\beta_1, \beta_2 \in [0,1)$. Then we have*

$$D_f(P \parallel Q) \leq \kappa^* D_g(P \parallel Q),$$

*where*

$$\kappa^* = \sup_{\beta\in(\beta_2,1)\cup(1,\beta_1^{-1})} \frac{f(\beta)}{g(\beta)}.$$

By Lemmas 2 and 6, we have

$$L(\lambda)H^2(P,Q) \leq D^\lambda_{\text{GJS}}(P \parallel Q) \leq U(\lambda)H^2(P,Q).$$

Now we show that $U(\lambda) \leq 1$. Its derivative $U'(\lambda)$ has a unique root at $\lambda = 1/2$ on the interval $(0,1)$ and it is positive if $\lambda < 1/2$ and negative if $\lambda > 1/2$. Therefore $U(\lambda) \leq U(1/2) = 1$.

### A.3  Proof of Lemma 1

The equation $m_\lambda(1) = 0$ can be verified by plugging in $t = 1$ directly. We compute the second derivative of $m_\lambda$

$$\frac{d^2 m_\lambda}{dt^2} = \frac{\lambda(1-\lambda)}{t^2\lambda + (1-\lambda)t}.$$

If $\lambda \in [0,1]$ and $t \in (0, \infty)$, we have $\frac{d^2 m_\lambda}{dt^2} \geq 0$, which implies the convexity of $m_\lambda$.

The $m_\lambda$-divergence equals to

$$D_{m_\lambda}(P \parallel Q) = \int_\Omega \lambda \ln \frac{dP}{dQ} dP - (\lambda dP + (1-\lambda)dQ) \ln \left( \lambda \frac{dP}{dQ} + 1 - \lambda \right)$$

while the MIL-divergence equals

$$\begin{aligned}
D_{\text{GJS}}^\lambda(P \parallel Q) &= \int_\Omega \lambda \ln \frac{dP/dQ}{\lambda dP/dQ + (1-\lambda)} dP + (1-\lambda) \ln \frac{1}{\lambda dP/dQ + (1-\lambda)} dQ \\
&= \int_\Omega \lambda \ln \frac{dP}{dQ} dP - (\lambda dP + (1-\lambda)dQ) \ln \left( \lambda \frac{dP}{dQ} + 1 - \lambda \right).
\end{aligned}$$

Thus we conclude that the $m_\lambda$-divergence yields the MIL-divergence with parameter $\lambda$.

### A.4  Proof of Proposition 1

Let $P$ and $Q$ be two probability measures in $\mathcal{P}$. If $P$ and $Q$ are equal, $D_f(P \parallel Q) = 0$. Therefore for any hash function $h$, it holds that $h(P) = h(Q)$, which implies that $\Pr_{h \sim \mathcal{H}}[h(P) = h(Q)] = 1 \geq p_1$.

In the sequel, we assume that $P$ and $Q$ are different. Since $P$ and $Q$ are two different distributions, there exists $i \in \Omega$ such that $P(i) < Q(i)$. We show this by contradiction. Assume that $\forall i \in \Omega$, $P(i) \geq Q(i)$. Since $P$ and $Q$ are different, there exists $i_0 \in \Omega$ such that $P(i_0) \neq Q(i_0)$. Since $P(i) \geq Q(i)$ holds for $\forall i \in \Omega$, we have $P(i_0) > Q(i_0)$. Therefore $\sum_{i \in \Omega} P(i) > \sum_{i \in \Omega} Q(i)$. However, both $P$ and $Q$ sum to 1, which leads to a contradiction. Therefore, we obtain the existence of $i$ such that $P(i) < Q(i)$, which yields $\beta_2 \triangleq \inf_{i \in \Omega} \frac{P(i)}{Q(i)} < 1$. Similarly, we have $\beta_1 \triangleq \inf_{i \in \Omega} \frac{Q(i)}{P(i)} < 1$. Since $P(i)$ and $Q(i)$ are non-negative for $\forall i \in \Omega$, we have $\beta_1, \beta_2 \geq 0$. In sum, we showed that $\beta_1, \beta_2 \in [0,1)$. By the definition of $\beta_0$, we know the following interval inclusion

$$(\beta_2, \beta_1^{-1}) \subseteq (\beta_0, \beta_0^{-1}).$$

Recall that

$$U = \sup_{\beta \in (\beta_0, 1) \cup (1, \beta_0^{-1})} \frac{f(\beta)}{g(\beta)},$$

$$L = \inf_{\beta \in (\beta_0, 1) \cup (1, \beta_0^{-1})} \frac{f(\beta)}{g(\beta)}.$$

By Lemma 6, we obtain the approximation guarantee

$$L \cdot D_g(P \parallel Q) \leq D_f(P \parallel Q) \leq U \cdot D_g(P \parallel Q) \tag{6}$$

There are two cases to consider. In the first case, we assume that $D_f(P \parallel Q) \leq Lr_1$. By (6), we have $D_g(P \parallel Q) \leq r_1$. Since $\mathcal{H}$ is an $(r_1, r_2, p_1, p_2)$-sensitive family for $g$-divergence, it holds that $\Pr_{h \sim \mathcal{H}}[h(P) = h(Q)] \geq p_1$. Similarly, if $D_g(P \parallel Q) > Ur_2$, we have $\Pr_{h \sim \mathcal{H}}[h(P) = h(Q)] \leq p_2$. Thus, $\mathcal{H}$ forms an $(Lr_1, Ur_2, p_1, p_2)$-sensitive family for $f$-divergence on $\mathcal{P}$.

## A.5   Proof of Theorem 2

If $D_{\mathrm{GJS}}^{\lambda}(P \parallel Q) \leq R$, by Theorem 1, we have

$$\left\| \sqrt{P} - \sqrt{Q} \right\|_2 \leq \sqrt{\frac{2R}{L(\lambda)}} \triangleq R_1.$$

If $D_{\mathrm{GJS}}^{\lambda}(P \parallel Q) \geq c^2 \frac{U(\lambda)}{L(\lambda)} R$, we have

$$\left\| \sqrt{P} - \sqrt{Q} \right\|_2 \geq c\sqrt{\frac{2R}{L(\lambda)}} = cR_1.$$

By the construction and properties of locality-sensitive hash family for $L^2$ distance proposed in [16, Section 3.2], we know that $h_{\mathbf{a},b}$ forms a $(R_1, cR_1, p_1, p_2)$-sensitive hash family for the $L^2$ distance between two vectors $\sqrt{P}$ and $\sqrt{Q}$. Therefore, provided that $D_{\mathrm{GJS}}^{\lambda}(P \parallel Q) \leq R$, which implies $\left\| \sqrt{P} - \sqrt{Q} \right\|_2 \leq R_1$, we have

$$\Pr[h_{\mathbf{a},b}(P) = h_{\mathbf{a},b}(Q)] \geq p_1.$$

Similarly, if $D_{\mathrm{GJS}}^{\lambda}(P \parallel Q) \geq c^2 \frac{U(\lambda)}{L(\lambda)} R$, we have

$$\Pr[h_{\mathbf{a},b}(P) = h_{\mathbf{a},b}(Q)] \leq p_2.$$

## A.6   Proof of Theorem 3

The derivative of the ratio function $\kappa(t) = \frac{\delta(t)}{\mathrm{hel}(t)}$ is

$$\kappa'(t) = \frac{1-t}{\sqrt{t}(t+1)^2}.$$

It is positive when $t < 1$ and negative when $t > 1$. Therefore for $\forall t \in (0, \infty)$, $\kappa(t) \leq \kappa(1) = 2$ and

$$\kappa(t) \geq \min\{ \lim_{t \to 0^+} \kappa(t), \lim_{t \to \infty} \kappa(t) \} = 1.$$

By Lemma 6, we have

$$H^2(P,Q) \leq \Delta(P \parallel Q) \leq 2H^2(P,Q).$$

If $\Delta(P \parallel Q) \leq R$, we have

$$\left\| \sqrt{P} - \sqrt{Q} \right\|_2 \leq \sqrt{2R} \triangleq R_1.$$

If $D_{\mathrm{GJS}}^{\lambda}(P \parallel Q) \geq 2c^2 R$, we have

$$\left\| \sqrt{P} - \sqrt{Q} \right\|_2 \geq \sqrt{2R}c = cR_1.$$

By the construction and properties of locality-sensitive hash family for $L^2$ distance proposed in [16, Section 3.2], we know that $h_{\mathbf{a},b}$ forms a $(R_1, cR_1, p_1, p_2)$-sensitive hash family for the $L^2$ distance between two vectors $\sqrt{P}$ and $\sqrt{Q}$. Therefore, provided that $\Delta(P \parallel Q) \leq R$, which implies $\left\| \sqrt{P} - \sqrt{Q} \right\|_2 \leq R_1$, we have

$$\Pr[h_{\mathbf{a},b}(P) = h_{\mathbf{a},b}(Q)] \geq p_1.$$

Similarly, if $\Delta(P \parallel Q) \geq 2c^2 R$, we have

$$\Pr[h_{\mathbf{a},b}(P) = h_{\mathbf{a},b}(Q)] \leq p_2.$$

## A.7  Proof of Lemma 3

First, we would like to note that $k$ is homogeneous, *i.e.*, for all $c \geq 0$, it holds that $k(cx, cy) = ck(x, y)$. Its kernel signature [35] is

$$\mathcal{K}(\lambda) \triangleq k(e^{\lambda/2}, e^{-\lambda/2}) = e^{-\frac{\lambda}{2}} \left( (e^{\lambda} + 1) \ln (e^{\lambda} + 1) - e^{\lambda} \lambda \right) .$$

First, let us review the definition of a positive definite function.

**Definition 4** ([9]). We call a complex-valued function $f : \mathbb{R} \to \mathbb{C}$ is positive definite if

  1. it is continuous in the finite region and is bounded on $\mathbb{R}$
  2. it is Hermitian, *i.e.*, $\overline{f(-x)} = f(x)$
  3. it satisfies the following conditions: for any real numbers $x_1, \ldots, x_n \in \mathbb{R}$, the matrix

$$A = (f(x_i - x_j))_{i,j=1}^n$$

  is positive semidefinite.

Next we will show that $\mathcal{K}$ is a positive definite function by showing that it is the Fourier transform of a non-negative function. We have the following Fourier transform and inverse Fourier transform

$$\mathcal{K}(\lambda) = \int_{\mathbb{R}} e^{-i\lambda w} \frac{2 \operatorname{sech}(\pi w)}{1 + 4w^2} dw ,$$

$$\kappa(w) \triangleq \frac{1}{2\pi} \int_{\mathbb{R}} \mathcal{K}(\lambda) e^{i\lambda w} d\lambda = \frac{2 \operatorname{sech}(\pi w)}{1 + 4w^2} .$$

Then we need the following lemmata.

**Lemma 7.** *If $f(x) = \int_{\mathbb{R}} e^{-ixt} g(t) dt$ is the Fourier transform of a non-negative function $g(t)$, then it is positive definite.*

*Proof of Lemma 7.* Let $x_1, \ldots, x_n \in \mathbb{R}$ be arbitrary real numbers and $a_1, \ldots, a_n$ be arbitrary complex numbers. Let us compute the quadratic form directly

$$\sum_{j,k=1}^n f(x_j - x_k) a_j \overline{a_k} = \int_{\mathbb{R}} \sum_{j,k=1}^n e^{-i(x_j - x_k)t} a_j \overline{a_k} g(t) dt = \int_{\mathbb{R}} \left| \sum_{j=1}^n a_j e^{-ix_j t} \right|^2 g(t) dt \geq 0 .$$

$\square$

**Lemma 8** (Lemma 1 in [35]). *A homogeneous kernel is positive definite if, and only if, its signature $\mathcal{K}(\lambda)$ is a positive definite function.*

Since $\frac{2 \operatorname{sech}(\pi w)}{1 + 4w^2} \geq 0$ holds for $\forall w \in \mathbb{R}$, we deduce that $\mathcal{K}(\lambda)$ is the Fourier transform of a non-negative function. Lemma 7 implies that $\mathcal{K}(\lambda)$ is a positive definite function. Therefore $k$ is a positive definite kernel by Lemma 8.

Let us define the feature map

$$\Phi_w(x) \triangleq e^{-iw \ln(x)} \sqrt{x \frac{2 \operatorname{sech}(\pi w)}{1 + 4w^2}} .$$

Since $k(x, y)$ is homogeneous, we have

$$k(x, y) = \sqrt{xy} k(\sqrt{x/y}, \sqrt{y/x}) = \sqrt{xy} \mathcal{K}(\ln(y/x))$$

$$= \sqrt{xy} \int_{\mathbb{R}} e^{-i \ln(y/x) w} \frac{2 \operatorname{sech}(\pi w)}{1 + 4w^2} dw = \int_{\mathbb{R}} \Phi_w(x)^* \Phi_w(y) dw .$$

## A.8  Proof of Theorem 4

Let $z$ denote the merged value. If we define $\eta(u) \triangleq -u\ln(u)$, the mutual information loss is

$$
\begin{aligned}
\text{mil}(\mathbf{x}, \mathbf{y}) &= \sum_{c\in\mathcal{C}}\left[p(c,x)\ln\frac{p(c,x)}{p(c)p(x)} + p(c,y)\ln\frac{p(c,y)}{p(c)p(y)} - p(c,z)\ln\frac{p(c,z)}{p(c)p(z)}\right]\\
&= \sum_{c\in\mathcal{C}}\left[p(c,x)\ln\frac{p(c,x)}{p(x)} + p(c,y)\ln\frac{p(c,y)}{p(y)} - p(c,z)\ln\frac{p(c,z)}{p(z)}\right]\\
&= \eta(p(x)) + \eta(p(y)) - \eta(p(z)) - \sum_{c\in\mathcal{C}}\left[\eta(p(c,x)) + \eta(p(c,y)) - \eta(p(c,z))\right] .
\end{aligned}
$$

By the definition of $k$, we have

$$
k(a,b) = \eta(a) + \eta(b) - \eta(a+b) .
$$

As a result, we re-write $\text{mil}(\mathbf{x}, \mathbf{y})$ as

$$
\text{mil}(\mathbf{x}, \mathbf{y}) = k(p(x), p(y)) - \sum_{c\in\mathcal{C}}k(p(c,x), p(c,y)) = K_1(\mathbf{x}, \mathbf{y}) - K_2(\mathbf{x}, \mathbf{y}) .
$$

Lemma 3 indicates that $k$ is a positive definite kernel. In light of the techniques for constructing new kernels presented in [8, Section 6.2], we obtain that that $K_1$ and $K_2$ are positive definite kernels.

## A.9  Proof of Lemma 4

Recall that $k(x,y) = \int_{\mathbb{R}}\Phi_w(x)^*\Phi_w(y)dw$. We have

$$
\begin{aligned}
\left|k(x,y) - \int_{-t}^{t}\Phi_w(x)^*\Phi_w(y)dw\right| &= \left|\int_{|w|>t}\Phi_w(x)^*\Phi_w(y)dw\right| \leq \int_{|w|>t}\left|e^{iw\ln(x/y)}\sqrt{xy}\rho(w)\right|dw\\
&\overset{(a)}{\leq} 2\int_{t}^{\infty}\rho(w)dw \overset{(b)}{\leq} 8\int_{t}^{\infty}e^{-\pi w}dw = \frac{8}{\pi}e^{-\pi t} \leq 4e^{-t} ,
\end{aligned}
$$

where $(a)$ is due to $\left|e^{iw\ln(x/y)}\sqrt{xy}\right| \leq 1$ and $(b)$ is due to

$$
\frac{2\,\text{sech}(\pi w)}{1+4w^2} \leq 2\,\text{sech}(\pi w) = \frac{4}{e^{\pi w}+e^{-\pi w}} \leq 4e^{-\pi w} .
$$

## A.10  Proof of Lemma 5

As the first step, we re-write the integral

$$
\int_{-\Delta J}^{\Delta J}\Phi_w(x)^*\Phi_w(y)dw = \sum_{j=-J+1}^{J}\int_{(j-1)\Delta}^{j\Delta}e^{iw\ln(x/y)}\sqrt{xy}\rho(w)dw .
$$

Then we bound the discretization error

$$
\begin{aligned}
&\left|\int_{-\Delta J}^{\Delta J}\Phi_w(x)^*\Phi_w(y)dw - \sum_{j=-J+1}^{J}\int_{(j-1)\Delta}^{j\Delta}e^{iw_j\ln(x/y)}\sqrt{xy}\rho(w)dw\right|\\
&\leq \sum_{j=-J+1}^{J}\int_{(j-1)\Delta}^{j\Delta}\left|e^{iw\ln(x/y)} - e^{iw_j\ln(x/y)}\right|\sqrt{xy}\rho(w)dw\\
&\overset{(a)}{\leq} \sum_{j=-J+1}^{J}\int_{(j-1)\Delta}^{j\Delta}|\ln(x/y)|\frac{\Delta}{2}\sqrt{xy}\rho(w)dw = \frac{\Delta}{2}\sqrt{xy}|\ln(x/y)|\int_{-\Delta J}^{\Delta J}\rho(w)dw \overset{(b)}{\leq} 2\Delta ,
\end{aligned}
$$

where $(a)$ is due to

$$
\left|e^{iw\ln(x/y)} - e^{iw_j\ln(x/y)}\right| \leq |\ln(x/y)||w - w_j| \leq \frac{\Delta}{2}|\ln(x/y)| .
$$

and (b) is due to $\int_{-\Delta J}^{\Delta J} \rho(w)dw \leq \int_{\mathbb{R}} \rho(w)dw = 2\ln 2$ and $\sqrt{xy}|\ln(x/y)| \leq \sqrt{x}|\ln(x)| + \sqrt{y}|\ln(y)| \leq \frac{4}{e}$.

Next we re-write the partial Riemann sum by substituting the new index $k = 1 - j$

$$\sum_{j=-J+1}^{0} \int_{(j-1)\Delta}^{j\Delta} e^{iw_j \ln(x/y)} \sqrt{xy}\rho(w)dw = \sum_{k=1}^{J} \int_{-k\Delta}^{(1-k)\Delta} e^{i(1/2-k)\Delta \ln(x/y)} \sqrt{xy}\rho(w)dw$$

$$= \sum_{k=1}^{J} \int_{(k-1)\Delta}^{k\Delta} e^{-iw_k \ln(x/y)} \sqrt{xy}\rho(w)dw \ .$$

Therefore the entire Riemann sum can be re-written as

$$\sum_{j=-J+1}^{J} \int_{(j-1)\Delta}^{j\Delta} e^{iw_j \ln(x/y)} \sqrt{xy}\rho(w)dw = \sum_{j=1}^{J} \int_{(j-1)\Delta}^{j\Delta} (e^{iw_j \ln(x/y)} + e^{-iw_j \ln(x/y)})\sqrt{xy}\rho(w)dw$$

$$= 2\sum_{j=1}^{J} (\cos(w_j \ln x)\cos(w_j \ln y) + \sin(w_j \ln x)\sin(w_j \ln y))\sqrt{xy}\int_{(j-1)\Delta}^{j\Delta} \rho(w)dw$$

$$= \left\langle \bigoplus_{j=1}^{J} \tau(x, w_j, j), \bigoplus_{j=1}^{J} \tau(y, w_j, j) \right\rangle \ .$$

## Appendix B    Illustration of Upper and Lower Bound Functions

We illustrate the upper and lower bound functions $U(\lambda)$ and $L(\lambda)$ in Fig. 3.

Figure 3: Upper and lower functions $U(\lambda)$ and $L(\lambda)$.

## Appendix C    Precision vs. Sketch Size

We show the precision vs. the sketch size in Fig. 4.

(a) Fashion MNIST          (b) MNIST          (c) CIFAR-10

Figure 4: Precision vs. speed-up factor for different sketch sizes.