[Reviews · NeurIPS 2019]

Reviewer 1



OVERVIEW. The paper presents locality-sensitive hashing schemes for well-studied distance function between probability distributions. The new schemes are based on the ideas. The first one is to approximate the distance function of interest by another distance function for which LSH schemes are known. In particular, the paper shows how to approximate MIL divergence and triangular discrimination by the Hellinger distance, for which LSH schemes are known. The second is specific to the MIL divergence, and involves representing the latter distance function as a so-called Krein kernel, and designing an asymmetric LSH scheme. The authors implement the last scheme and evaluate its performance in the context of approximate nearest neighbor search, showing significant speedup over linear scan, with a reasonable precision. EVALUATION. The problem addressed by the paper (similarity search for probability distributions) is interesting, relevant and well-studied. The proposed solutions are interesting, even if not terribly deep: the proofs involve lots of tedious but conceptually straightforward calculations. They also seem to work well in practice. One issue is that the data sets (e.g., MNIST) used for the evaluation seem somewhat artificial in the context of probability distributions. Specifically, an image is not really a probability distribution, although it can be viewed as such. Nevertheless, the overall paper adds important insights to the similarity search literature. MINOR COMMENTS: - There are multiple papers on this topic not listed in the paper, see below. The authors might want to discuss them in the related work section. Abdullah, Amirali, John Moeller, and Suresh Venkatasubramanian. "Approximate Bregman near neighbors in sublinear time: Beyond the triangle inequality." Proceedings of the twenty-eighth annual symposium on Computational geometry. ACM, 2012. Abdullah, Amirali, and Suresh Venkatasubramanian. "A directed isoperimetric inequality with application to bregman near neighbor lower bounds." Proceedings of the forty-seventh annual ACM symposium on Theory of computing. ACM, 2015. Abdelkader, A., Arya, S., da Fonseca, G. D., & Mount, D. M. (2019). Approximate nearest neighbor searching with non-Euclidean and weighted distances. In Proceedings of the Thirtieth Annual ACM-SIAM Symposium on Discrete Algorithms (pp. 355-372). Society for Industrial and Applied Mathematics. - The author measure the improvement of their scheme over the linear scan by the ratio of wall clock runtimes. Although useful, such measurements can be affected by the implementation details, specific computer architecture etc. It would be helpful if the author included more "combinatorial" measures such as the number of points examined by the LSH algorithm during the query processing stage

Reviewer 2



originality: the paper proposes a new method for obtaining LSH schemes for distances over distributions. clarity: the paper is relatively easy to follow, with proofs in the appendix. There are some minor suggestions for improving the clarity (see comments below) significance: LSH is a popular technique, and adding new similarity functions for which there are LSH schemes is useful for others. There are some issues that put the usability of the new schemes in question, see below. Major comments: Theorem 2: the guarantees of the new LSH scheme depends on the constant c - the distance between two specific distributions P and Q. However, the LSH scheme guarantees are for the entire family of distributions (the 'universal set of items'). This should be fixed, possibly by replacing c with an upper bound on the distance between *any* pair of distributions P,Q, with an appropriate discussion as to whether this is a reasonable constant (i.e. will a value of c that is too large make this LSH scheme useless). Sec. 4, Krein-LSH: it is not clear what the motivation is for this section. Are the guarantees for MIL-LSH based on the reduction from squared Hellinger distance not good enough? are the two methods comparable? Sec. 5, "nearest neighbor search": which MIL LSH scheme is used? the reduction from squared Hellinger or the Krein kernel? What were the obtained guarantees? How do these affect the correctness / efficiency tradeoff? Do they seem to be tight? Sec. 5: it is odd that for the "well-motivated" mutual information loss, the chosen datasets of distributions are artificially generated from image datasets. Why would the MIL be a reasonable loss to consider for images? Other comments, in order of appearance: lines 56-67: the wording is somewhat redundant. Also, the "tool for designing LSH schemes for the family of f-divergence distance functions" can only be applied as a reduction tool, i.e. it is required that there will be an existing LSH scheme for a related f-divergence. line 63: it is claimed that the MIL is "well motivated", but this motivation is only described later in lines 69-71. Sec. 2.1: when presenting LSH for the first time, it is worthwhile to define $U$, the set of possible hash values, as well as discuss its size and the resulting trade-offs. Sec. 2.2: There is no reference to any Information Theory literature (e.g Cover and Thomas / Gallager), e.g. for the definitions of mutual information, KL divergence, Shannon entropy line 138: the citation format is off - it should include the authors within the citation End of Sec. 2: it is worthwhile to mention that the MIL (and, as a result, the JSD) are both f-divergences, and that this fact will be shown later on. It is not clear from the current wording. Prop. 1: in the statement of the proposition, \beta_0, L and U should be formally stated (e.g. 'let \beta_0>0, ..' etc.). Line 205: the notation is a bit confusing at first. It might help to define the function p() with a different variable than c (that is also used as a constant). line 209: the term 'sketched' is used here for the first time. Please define it. Sec. 4: it would help to cite a general source about Krein kernels. line 229: confusing definitions for the function \eta(x), since $x$ is a data point Sec. 4.1: it would help to explicitly define p(z) and p(c,z). line 238: the term "assymetric LSH" is used here for the first time. Please define it. Figure 2: it would help to also use dashed / dotted lines in order to make the figure legible in black and white printing Appendix A.6, proof of Lemma 5: there is a redundant $x$ in the exponent (twice) After reading the rebuttal letter: please indicate that the reason for choosing artificially generated distributions is because it was done in previous papers. However, it would be better to add an evaluation on a dataset where the data points are more real-life distributions. Also, the comparison between Hellinger-LSH and Krein-LSH (for the MIL case) should be analyzed in detail. the losslessness property of the Krein method is not clear from the paper. It would be best if there is a separate section that compares both methods. If the Krein-LSH is always better, state it clearly.

Reviewer 3



#### EDIT after feedback #### I have read the feedback. Comparison with L^2 baseline would be a nice addition. The response also helped understand the role of Section 4. ######################### This paper presents LSH schemes where each data point is regarded as a probability distribution (on a discrete random value) and measured by f-divergences. The LSH for a certain f-divergence is approximated by another f-divergence that gives an LSH family. In concrete, the paper bounds several divergences with the squared Hellinger distance which gives the L^2-LSH. My comments and suggestions are * Experiments It is nice to see the empirical validation of the approximation guarantee as well as the performance of k-nearest search. My concern is that the comparison with existing LSH families lacks in Figure 2. L^2 LSH would be a reasonable baseline. * Section 4 This part is rather independent of the rest of the paper. While mutual information loss is motivated for compression tasks, the described algorithm is not readily connected with that task. Experiments are not presented either. What is the strength of this hashing scheme? * Proposition 1 While Proposition 1 constitutes the core to develop LSH schemes for several f-divergences, the claim is tough to follow at first glance. If possible, some visualization helps the understanding. For example, how the bound L \leq f/g \leq U connects (r1, r2, p1, p2)-sensitive family with (Lr1, Ur2, p1, p2) one.

[Author Response · NeurIPS 2019]

We thank the reviewers for their comments. We will fix all minor issues and do not discuss them individually here.

**Common Questions:**

**Q1:** Use datasets with natural probability distributions?

**A1:** We appreciate your concern and will consider other datasets as well. However, we would like to remark that
we used the same distributions used in Mao et al., 2017, which is the most relevant work. Mao et al. studies LSH
of Jensen-Shannon distance (an information-theoretic distance) and normalizes the images in MNIST and CIFAR to
probability distributions.

**Q2:** Krein-LSH (Sec. 4) vs. the Hellinger-approximation-based LSH (Sec. 3).

**A2:** Our Hellinger-approximation-based LSH is a more general method and applies to the family of $f$-divergences
(including MIL). However, particularly focusing on MIL, Krein-LSH has a better approximation factor. In fact, our
Krein-LSH is *lossless*, i.e., we do not lose anything on the approximation factor except for computing the integral
numerically. We have a more detailed discussion in lines 68–75.

**Response to Reviewer #1:**

**Q3:** Data sets seem somewhat artificial in the context of probability distributions. **A3:** Please see **A1**.

**Q4:**- There are multiple papers on this topic not listed in the paper, see below.

**A4:** Thank you for providing the references. We will discuss and compare them in the related work section.

**Q5:** It would be helpful if the author included more "combinatorial" measures such as the number of examined points.

**A5:** Thank you for this great suggestion. We will definitely report the number of examined points in our final version.

**Response to Reviewer #2:**

**Q6:** Theorem 2: $c$ should be fixed, possibly by replacing $c$ with an upper bound. Is it a reasonable constant?

**A6:** Thanks for noting this. We agree with you. It is possible to drive an upper bound of $c \leq \sqrt{2}$. This is serves its
purpose since one can always increase the code length and the number of hash functions (buckets) to separate close and
faraway points apart. We will address this in the paper.

**Q7:** Sec. 4, Krein-LSH: it is not clear what the motivation is for this section. Are the guarantees for MIL-LSH based
on the reduction from squared Hellinger distance not good enough? are the two methods comparable?

**A7:** Please see **A2** for comparison between MIL-LSH and Krein-LSH. We will make the motivation clearer.

**Q8:** Sec. 5: which MIL LSH is used? What were the Obtained guarantees? How do these affect the correctness-
efficiency tradeoff? Do they seem to be tight?

**A8:** We used the MIL-LSH via Hellinger approximation here. To show the guarantees, we evaluate the performance
by solving an end-to-end nearest neighbor search problem (this is a standard setup to evaluate LSH methods). All six
subfigures in Fig. 2 illustrate the correctness-efficiency tradeoff. All figures seem as expected.

**Q9:** Sec. 5: the chosen datasets of distributions are artificially generated from image datasets. **A9:** Please see **A1**.

**Response to Reviewer #3:**

**Q10:** $L^2$ LSH would be a reasonable baseline.

**A10:** Thank you for the constructive comment. We have results of $L^2$-LSH. On MNIST, when its speedup factor is 2,
the precision is only 0.7 (ours is 0.85). When its speedup factor is 3, the precision drops below 0.63 (ours is 0.73).
Our method clearly outperforms it. Due to the space limit here, we will present them in the final version.

**Q11:** Section 4 is independent of the rest. The described algorithm is not readily connected with that task. Experiments
are not presented either. What is the strength of this hashing scheme?

**A11:** Sec. 4 is indeed an improvement for the method described in Sec. 3. For the strength of Krein-LSH, please refer
to **A2**. The described algorithm reduces the problem of finding small MIL to the problem of maximum inner product
search and uses, say, SIMPLE-LSH (Neyshabur and Srebro, 2014) at the final stage (Line 5 of Algorithm 1). We will
clarify these points in the final version. We have experiment results regarding Krein-LSH and will include them in the
final version by moving some other content to the appendix.

**Q12:** Proposition 1 is tough to follow at first glance. If possible, some visualization helps the understanding.

**A12:** Thank you for this valuable comment. We will provide a proof sketch for this key proposition.

[Meta-Review · NeurIPS 2019]

There were certain disagreement initially about the paper. However, after rebuttal and post discussion, all reviewers agreed that the contributions and methodology presented in the paper are interesting enough and deserves to be published. Authors are encouraged to take into account the reviews before preparing the final version of the paper.